# Language and Communication Interventions in People with Alzheimer’s Disease: A Systematic Review

**DOI:** 10.3390/healthcare12070741

**Published:** 2024-03-29

**Authors:** Nefeli K. Dimitriou, Anastasia Nousia, Eleni-Nefeli Georgopoulou, Maria Martzoukou, Ioannis Liampas, Efthimios Dardiotis, Grigorios Nasios

**Affiliations:** 1Department of Speech and Language Therapy, School of Health Sciences, University of Ioannina, 45500 Ioannina, Greece; nefdimitriou@uth.gr (N.K.D.); e.georgopoulou@uoi.gr (E.-N.G.); grigoriosnasios@gmail.com (G.N.); 2Department of Speech and Language Therapy, University of Peloponnese, 24100 Kalamata, Greece; a.nousia@go.uop.gr; 3Lab of Cognitive Neuroscience, School of Psychology, Aristotle University of Thessaloniki, 54124 Thessaloniki, Greece; mmartzo@enl.auth.gr; 4Department of Neurology, University Hospital of Larissa, Faculty of Medicine, School of Health Sciences, University of Thessaly, 41100 Larissa, Greece; edar@med.uth.gr

**Keywords:** language training, communication rehabilitation, Alzheimer’s disease

## Abstract

Although language impairment is frequently observed in patients with Alzheimer’s disease (pwAD), targeted language rehabilitation is often overlooked. The present study reviews published evidence on the impact of language training, either alone or in combination with cognitive training, on cognitive outcomes in pwAD. A systematic search of PubMed, Google Scholar, and Cochrane was carried out, including studies published from inception to November 2023. A total of eight research articles (four randomized controlled trials and four observational studies) met the inclusion criteria: six assessed language training combined with cognitive training and two evaluated language rehabilitation alone. Regarding language and non-language (mainly memory, attention, and executive functions) outcomes, there was a consensus among studies that language rehabilitation (alone or in combination with cognitive training) yields positive results. Some of the articles also explored the impact on patients’ and their caregivers’ quality of life, with all but one showing improvement. Consequently, the combination of language and cognitive training leads to improvements across various cognitive domains. However, limited evidence supports the value of sole language rehabilitation. This conclusion is influenced by heterogeneity among studies (different types and duration of interventions, small participant sets, various assessment tools), and, thus, further research is warranted.

## 1. Introduction

Alzheimer’s disease (AD) and other major neurocognitive disorders represent important public health concerns. According to the National Institute of Neurological and Communicative Disorders and Stroke and Alzheimer’s Disease and Related Disorders Association (NINCDS ADRDA) [1] and the Diagnostic and Statistical Manual of Mental Disorders (DSM-V) [2] criteria, AD diagnosis cannot be determined by laboratory tests; structural, and functional imaging techniques, along with clinical and neuropsychological testing, are of utmost importance. The clinical picture includes progressive impairment of memory - other cognitive and language functions [3].

In greater detail, non-language - cognitive manifestations include deterioration of episodic memory, semantic memory, working memory, executive function, and visuospatial skills [4,5]. Of note, language impairment is often underestimated [6]. Nevertheless, language deficits occur early in AD compared to other cognitive domains, and performance on verbal fluency tasks serves as an important screening procedure and diagnostic criterion [7,8]. This language decline appears to be hierarchical in nature, with the language forms learned last in the sequence of language development deteriorating first [9]. Early stages are characterized by anomia [10], repetitions, and periphrasis; aphasic-like language occurs at later disease stages [11]. Semantic errors (e.g., difficulty finding the right word, poor vocabulary) are frequent [5,12], and semantic [13] and letter fluency [14] problems can be identified. The domain of syntax consists of acceptable sentences regarding simple, common, and automated forms, but grammatical and syntactic difficulties emerge in more complex structures at early stages [12,15]; sentence comprehension deficits vary with the degree of dementia severity [16]. At mild-to-moderate stages, morphosyntactic production and comprehension impairments are evident both on word [17,18] and sentence levels [19,20]. Finally, phonology is the most well-preserved language domain in AD [9], although the production of phonemic paraphasias and neologisms is not rare [21].

Neuropsychological assessment plays a critical role in characterizing cognitive and language deficits associated with AD [22]. Language is a multiscale system that we use to decode symbols (word, sign, or other forms of linguistic labels) to convey or comprehend a message, and to apply the appropriate grammatical and syntactical rules. However, the literature shows a close relationship between language brain networks and other aspects of cognition, suggesting a difficulty in assessing and treating it in isolation [23]. Clinicians can evaluate the elements of language function via conversational and spontaneous speech, naming exercises, comprehension tasks (such as giving specific commands to the patient), repetition, reading, and writing [24].

To date, many pharmacological therapies have been explored but have failed to achieve satisfactory outcomes regarding language and cognitive impairments [25]. Non-pharmacological strategies, however, have been found effective in improving or at least maintaining the concurrent level of handicap [26,27]. In greater detail, cognitive rehabilitation (including activities for language improvement), physical exercise, music therapy, behavioral and psychological interventions, occupational therapy, complementary and alternative medicine (such as acupuncture therapy), and new technologies (non-invasive brain stimulation, assistive technology and domotics, virtual reality, gaming, and telemedicine) have demonstrated respectable results [28,29].

It appears that most researchers have focused on cognitive interventions, which may or may not include language rehabilitation, and have involved patients with various types of dementia [30,31,32,33]. Notably, the most structured cognitive therapy programs emphasize on memory [34].There is limited understanding of interventions specifically aiming to improve language and communication deficits in people with Alzheimer’s disease (pwAD).

In view of the aforementioned gaps in the literature, we conducted a systematic review to summarize evidence from studies implementing language and communication interventions for pwAD. In particular, the present systematic review aimed to (a) explore the effectiveness of language and communication interventions on pwAD and (b) determine which language and cognitive skills benefit more from these interventions.

## 2. Materials and Methods

The current report followed the PRISMA guidelines for reporting systematic reviews (Appendix A) [35]. It was not pre-registered on an online database prior to its conduction. Each step of the process (literature search, study selection, data extraction, quality evaluation) was performed by two authors (N.D. and A.N.). Discrepancies were resolved by a third author (I.L.). Inter-rater reliability was not statistically established due to the lack of an exact number of items assessed—it was determined only with respect to quality evaluation. Among the 20 in total assessed items (5 per study, 4 studies) only one discrepancy was resolved by the third author. Nineteen in twenty items (95%) were scored similarly.

### 2.1. Search Strategy

An extensive literature search was conducted in PubMed, Google Scholar, and CENTRAL. The following search terms were used (entered as free words): (a) “language training” and “Alzheimer” and (b) “Alzheimer” and “language rehabilitation” and “communication training”. The final literature search was performed on the 30 November 2023. The initial search yielded a total of 1.538 studies published up to 30 November 2023. In specific, 1296 studies were derived from PubMed, 120 studies from Google Scholar (search terms only in title), and 149 studies from CENTRAL.

### 2.2. Data Extraction

The following data were extracted from the retrieved studies: author, year of publication, number of participants, AD stage, targeted domains through the intervention, study design, outcome measures, duration and frequency of the intervention, and outcomes.

### 2.3. Eligibility Criteria

The following eligibility criteria were considered: (1) use of language and communication training, (2) inclusion of participants with AD -exclusively, (3) availability of pre- and post-intervention cognitive data, and (4) inclusion of at least 5 participants in the language rehabilitation arm. Studies were excluded if they: (1) lacked pre- or post-intervention cognitive data, (2) involved different kinds of interventions (e.g., physical training, cognitive training without language training, and so on), (3) included participants with other neurological conditions (e.g., stroke, Parkinson’s), (4) were not original research articles (e.g., review articles, meta-analyses). Furthermore, language restriction criteria were applied; only articles published in English were considered for eligibility. Conclusions were based on randomized controlled studies (RCTs). Due to the small number of retrieved articles, the findings of observational studies are also narrated. Further information on the procedure of selection of eligible studies is presented in Figure 1 [35].

### 2.4. Risk of Bias Assessment Tool

Risk of bias was assessed using the Risk of Bias (RoB) Cochrane tool for Systematic Reviews of interventions. Five methodological domains were appraised: (1) randomization process (allocation sequence generation, allocation concealment, baseline between group differences), (2) deviations from intended interventions (blinding of participants, blinding of personnel, appropriate analysis), (3) missing outcome data (data availability, reasons for missing data), (4) measurement of outcomes (method of measurement for both groups) and (5) selective reporting (prespecified protocol, multiple analyses). Each item was rated as of “low risk of bias”, “high risk of bias”, or “some concerns” based on methodological features and reporting of the retrieved studies (Figure 2).

## 3. Results

### 3.1. Study Characteristics

Study characteristics are in Table 1. One study was conducted in France [36], one in United Kingdom (UK) [40], two in Italy [37,41], one in Spain [42], one in Greece [38], one in India [43], and one in Turkey [39]. Sample sizes ranged from 8 to 80 participants: in total, 272 AD patients were included, 183 of which were in the “training group” (TG) and 89 were in the “control group” (CG).

### 3.2. Patients’ Characteristics

The patients’ characteristics are in Table 2. TG included 183 individuals: 70 males and 105 females, while CG consisted of 89 participants: 30 males and 68 females [36,37,38,39,41,42,43]. The study by Noonan et al. [40] did not provide any information about the gender of the participants.

The age of the participants ranged from 58 to 92 years old. There were no significant differences in age and gender between the groups [36,37,38,39,40,41,43], except for the study conducted by Martinez-Moreno et al. [42]. Moreover, the formal education of the participants varied from 2 to ≥13 years.

Six studies used the National Institute of Neurological and Communicative Disorders and Stroke and the Alzheimer’s Disease and Related Disorders Association (NINCDS/ADRDA) criteria [36,38,40,41,42,43]. Cavallo et al. [37] provided no further information about the diagnostic process; in the study of Parlak et al. [39], the Diagnostic and Statistical Manual of Mental Disorders 5th edition (DSM-5) and the Nationals’ Institute on Aging and Alzheimer’s Association (NIA-AA) criteria were fulfilled. The AD stage of the participants varied from mild to severe.

### 3.3. Intervention Characteristics

In all studies, participants underwent training programs that lasted for periods ranging from 5 to 48 weeks, with a frequency of sessions varying from 1 to 5 times per week (Table 3). The duration of each session ranged from 30 min to 2 h. Specifically, in the study of Bajpai et al. [43], participants had daily sessions (5 days/week) lasting 30–45 min, with a total intervention period of 8 weeks. In the study of Spironelli et al. [41], there were 4 sessions/week, lasting a total of 2 h, along with additional practice activities of daily living at home for a total of 5 weeks. Nousia et al. [38] included 2 sessions per week, with a total duration of 15 weeks, consisting of 60 min of multidomain intervention. Cavallo et al.’s [37] intervention consisted of 3 weekly 30 min sessions over a period of 12 weeks. Parlak et al. [39] conducted five 60-min sessions/week for 7 weeks. Martinez-Moreno et al.’s [42] study implemented a 1-year group program, with 2–3 weekly two-to-three-hour sessions. Lastly, Ousset et al. [36] provided 16 sessions, once a week, over a period of 5 months, lasting 45 min each, with a 2-week break in intervention between the 8th and 12th week.

The participants performed the sessions either individually [36,37,38,39,41,43], within groups [42], or both [38]. In 4 out of 8 studies, tasks were developed and performed with computer aid [36,37,39,41]. Two studies used paper-pencil training [42,43]. In the study of Nousia et al. [38], the intervention was performed using both computer aid and paper-pencil means, while Noonan et al. [40] did not provide further information about the means of training. In the study of Bajpai et al. [43], caregivers were trained to perform the training using paper and pencil.

### 3.4. Language and Cognitive Domains Targeted

Most of the studies focus on the training of more than one cognitive domain [37,38,39,41,42,43]. Specifically, apart from language, the main functions these studies tried to enhance were memory, attention, and executive functions. There are two studies, however, that solely targeted language [36,40]. In particular, Noonan et al. [40] aimed to improve patients’ name relearning, and Ousset et al. [36] conducted a lexical therapy with naming sessions.

Regarding outcome assessment, participants in five studies [37,38,40,41,42] completed screening cognitive tests (Mini-Mental State Examination, Trail-making test A and B, digit forward and backward span, etc.). The language domain was assessed through naming tasks (such as the Boston Naming Test) and semantic or phonological fluency tasks. Only one study [39] included a more detailed language evaluation; the Language Assessment Test for Aphasia (LATA) investigated speech fluency, auditory comprehension, repetition, naming, reading, grammar, word actions, and writing. Although most studies based the reported improvement in patients’ performance on specific neuropsychological tests, Bajpai et al. [43] measured the difference in reaction time before and after intervention in order to infer the positive effect of training on memory, attention, and language domains. Similarly, Ousset et al. [36] measured participants’ naming hits and errors. Detailed outcomes are provided in Table 4.

Four out of eight studies [36,37,38,39] can be characterized as randomized controlled trials (RCT). In the study of Ousset et al. [36], seven out of eight patients in the study group demonstrated improved performance on naming post-therapy. The second RCT [37] revealed that the training group performed significantly better on the digit span-forward, digit span-backward, two-syllable words repetition test, Rivermead Behavioral Memory Test (RBMT)-story immediate, RBMT-story delayed, Token test and Brixton test compared to the control group. These improvements were sustained for at least 6 months after training. The third RCT [38] demonstrated better results for the patients who underwent the intervention in all cognitive domains. Specifically, improvements were noted in delayed memory, visuospatial abilities, executive functions, working memory, naming, semantic fluency, and attention/processing speed, whereas milder improvements in recall and recognition was also noticed. Finally, Parlak et al. [39] observed that their intervention was effective in improving speech fluency and auditory comprehension for mild and moderate AD, grammar for all stages, and repetition and speech act skills for moderate AD. In addition, the mean Language Assessment Test for Aphasia (LATA) and Mini Mental State Examination (MMSE) scores of the study group were increased.

The four observational studies [40,41,42,43] provide supporting evidence regarding the effectiveness of language and cognitive training, as well. More specifically, Bajpai et al. [43] demonstrated improvement in practiced tasks, especially in the domains of episodic and semantic memory. In the study by Martínez-Moreno et al. [42], 51.7% of patients classified as responders exhibited better performance on global cognitive performance, orientation, and executive function compared to non-responders, whereas all participants showed improvement in spatial orientation abilities. Noonan et al. [40] showed that both errorless and errorful learning can benefit the naming ability of pwAD. On the contrary, participants in Spironelli et al.’s [41] study did not show significant changes in neuropsychological tests after cognitive training.

Only two studies [37,40] investigated the duration of the training outcome by exploring its effects five weeks and six months after the completion of the intervention, respectively. Noonan et al. (2012) [40] found only borderline significant correlations between overall relearning scores and the 100-item naming test, the 64-item word–picture matching task and Elevator Counting with Distraction from the Test of Everyday Attention. On the contrary, Cavallo et al. (2016) [37] concluded that patients in TG showed a better performance in comparison to CG on digit span-forward, digit span-backward, two-syllable words repetition test, RBMT-story immediate, RBMT-story delayed, Token test and Brixton test, exactly as observed at the post-treatment assessment.

### 3.5. Noncognitive Outcomes: Quality of Life Outcome Measures

More than half of the aforementioned studies, in addition to assessing language and cognitive domains, also evaluated the potential impact of training on patients’ quality of life and psychological status. Two studies [37,42] used specialized tests, such as the Instrumental Activities of Daily Living (IADL) and the Hospital Anxiety and Depression Scale (HADS), whereas the other three [38,39,41] relied on subjective statements from caregivers (Table 3).

Among these studies, three were RCT [37,38,39]. The latter two [38,39] revealed promising outcomes based on patients’ verbal feedback after the completion of the intervention. On the contrary, Cavallo et al. [37] compared patients’ scores on the HADS before and after treatment and did not find any statistically significant difference. In the other two studies, participants described as responders in Martínez-Moreno et al.’s study [42] showed a better performance in IADL after treatment, whereas in Spironelli et al.’s study [41] treatment helped participants to develop a sense of support towards those with more severe deficits.

## 4. Discussion

The enhancement of patients’ language skills is vital, given that the loss or impairment of communication ability affects not only patients but also their caregivers and relatives [26,44], leading to a profound deterioration in patients’ quality of life [45]. Apart from the psychological burden, families and caregivers have to contend with enormous healthcare expenses [46]. The yearly worldwide cost of dementia is estimated at $1.3 trillion [47]. Thus, accurate and prompt intervention is of utmost importance [48,49].

The examination of the retrieved studies revealed considerable benefits from language rehabilitation alone, as well as in combination with other cognitive tasks. The studies that showed greater improvements were those that implemented holistic language-cognitive intervention programs. In particular, training groups in the studies conducted by Cavallo et al. [37], Martínez-Moreno et al. [42], Nousia et al. [38], and Parlak et al. [39] demonstrated better outcomes on various assessment tools evaluating both language and cognitive domains. The other two surveys [41,43] showed fewer substantial results; the first one concluded that pwAD demonstrated improved mean time in the tasks, and the second one found that patients’ performance improved significantly only in abstraction and phonemic fluency subtests. In addition, an interaction effect appears to be present in the study by Bajpai et al. [43]. On the other hand, two of the studies that applied pure language intervention [36,40] examined the performance solely on naming tasks. Of note, only four studies were RCTs [36,37,38,39]. Of these, only one [36] consisted of pure language treatment, whereas the others included both language and cognitive interventions. As a result, it is risky to draw conclusions based only on the RCTs.

To the best of our knowledge, the present review is the first to focus on the impact of specific language and communication interventions on the overall language profile, cognitive status, and quality of life of pwAD. Our findings are in agreement with those of Morello et al.’s study [50], which examined the effects of non-pharmacological interventions in pwAD and found that lexical-semantic approaches and treatments targeting various cognitive domains (including language) appear effective. This study, however, does not exclusively focus on language restoration but it also includes interventions that combine language with physical activities, various conversational techniques, and communicative training of caregivers. Other systematic reviews have investigated the effect of cognitive rehabilitation, which also includes language interventions in some cases, but they do not exclusively involve pwAD. Instead, the population in these studies consisted of pwAD and other types of dementia or mild cognitive impairment (MCI) [30,33,51,52,53].

The limited evidence regarding exclusive language rehabilitation may result from the fact that most studies do not focus on language interventions and do not compare language with global cognitive intervention. Future studies should be conducted with a focus on exclusive language rehabilitation and the comparison between language and cognitive treatment techniques.

Additional limitations need to be acknowledged. These limitations stem, mainly, from the heterogeneity among studies. A methodological limitation is the fact that the participants in all the included surveys, except for one, were patients with mild AD. Future studies should organize the presentation of their findings based on dementia type (i.e., Alzheimer’s disease, vascular dementia, etc.) and the severity of the disease (i.e., mild, moderate, or severe) to reflect the clinical importance of potential benefits at each stage of severity. In addition, future surveys should give prominence to language interventions with longer durations and follow-up evaluations. Finally, some selected articles included only a few patients, and none of them adhered to an optimum methodological quality with blinding procedures.

Moreover, significant inconsistencies are apparent regarding the evaluation tools used in the analyzed articles. Language evaluation should be conducted with batteries that have good psychometric properties and can investigate various aspects of language and communication difficulties that pwAD experience frequently. According to recent data [54] such tools are the Arizona Battery for Cognitive-Communication Disorders (ABCD) [55], the Sydney Language Battery (SydBat) [56], and the Addenbrooke’s Cognitive Examination (ACE-III) [57]. Nevertheless, this review includes studies that evaluate language outcomes with other tools that assess various cognitive domains, including language in general. Moreover, as discussed above, language evaluation cannot completely exclude other cognitive domains, such as memory, due to the strong connection between language and cognitive brain networks [23].

## 5. Conclusions

In conclusion, the present systematic review revealed a significant positive effect of the combination of language and cognitive intervention in both language and cognitive domains. On the other hand, pure language interventions showed improvement only in language tasks. In respect to the possible improvement of the quality of patients’ life, the evidence are weak or subjective, often relying on statements from patients’ caregivers.

Considering these findings, the inclusion of language and communication training (either alone or in combination with cognitive training) is recommended for healthcare practitioners to enhance their patients’ neuropsychological function, thereby improving their quality of life.

## Figures and Tables

**Figure 1 healthcare-12-00741-f001:**
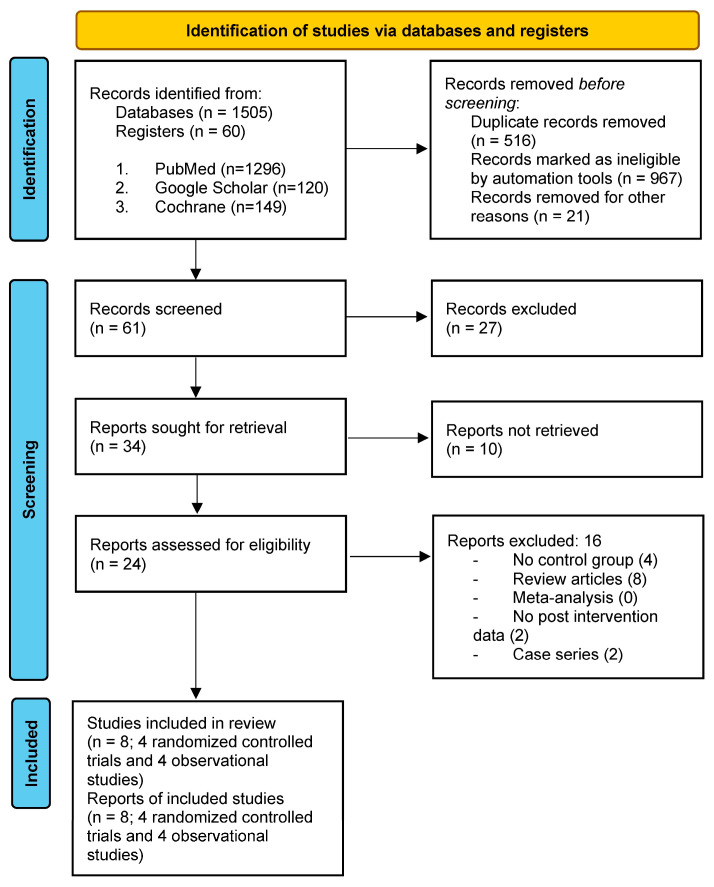
Prisma 2020 Flow Diagram.

**Figure 2 healthcare-12-00741-f002:**
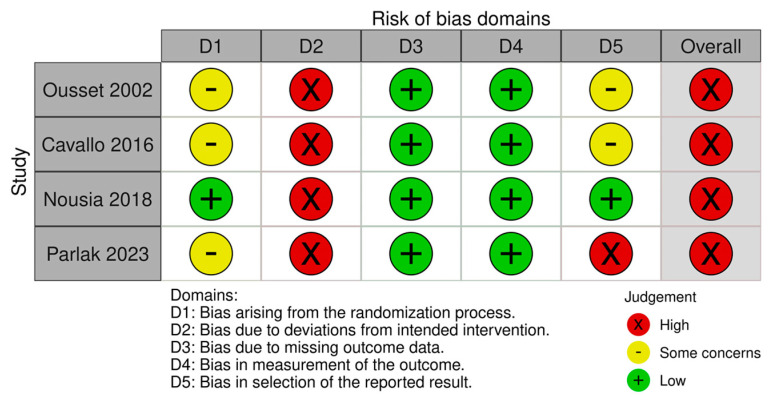
Risk of bias (RoB) of included studies [36,37,38,39].

**Table 1 healthcare-12-00741-t001:** General information of the retrieved studies.

Study	Country	Groups	Participants per Group	Intervention
Ousset et al. (2002) [36]	France	AD ^1^	8 LT ^2^ group 8 OT ^3^ group	(LT ^2^) Naming sessions presented on a computer
Noonan et al. (2012) [40]	United Kingdtom	AD ^1^	8	Naming sessions
Spitonelli et al. (2013) [41]	Italy	AD ^1^	11	Tasks were presented by either paper and pencil or by a computer
Cavallo et al. (2016) [37]	Italy	AD ^1^	40 TG ^4^ 40 CG ^5^	Rehabilitative software Brainer1
Martínez-Moreno et al. (2016) [42]	Spain	AD ^1^	60	At the Outpatients Clinics in the Day Hospital (pen and paper tasks)
Nousia et al. (2018) [38]	Greece	AD ^1^	25 TG ^4^ 25 CG ^5^	1st part = 30 min. computer-based intervention 2nd part = 30 min. exercises with paper and pencil
Bajpai et al. (2020) [43]	India	AD ^1^	15	AD ^1^: tasks with a trained caregiver
Parlak et al. (2023) [39]	Turkey	AD ^1^	16 TG ^4^ 16 CG ^5^	Computer-supported application (software) at home

^1^ AD = Alzheimer’s disease; ^2^ LT = Lexical Therapy; ^3^ OT = Occupational Therapy, ^4^ TG = training group; ^5^ CG = control group.

**Table 2 healthcare-12-00741-t002:** Demographics of study participants of the retrieved studies.

Study	N ^1^	AD ^2^ Type	Diagnostic Criteria	CDR ^3^	GDS ^4^ Mean (±SD)	Gender (Males/Total)	Age in Years [Mean (SD ^5^)]	Education in Years [Mean (SD ^5^)]	Pharmacological Treatment
Ousset et al. (2002) [36]	8 AD ^2^ (LT ^6^)	Probable AD ^2^	NINCDS-ADRDA ^8^	-	-	5/8	67.7 ± 12.9	-	Cholinergic medication
	8 AD ^2^ (OT ^7^)					3/8	73.8 ± 7.5	-	
Noonan et al. (2012) [40]	8 AD ^2^	Probable AD ^2^	NINCDS-ADRDA ^8^	-	-	-	-	-	-
Spironelli et al. (2013) [41]	11 AD ^2^	Mild-to-moderate AD ^2^	NINCDS-ADRDA ^8^	-	-	2/11	78.18 (±4.99) Range = 70–88	7.54 (±3.59)	Anticolinesterasic drugs
Cavallo et al. (2016) [37]	40 AD ^2^ (TG ^9^)	Early stage probable AD ^2^	NINCDS-ADRDA ^8^	-	-	13/40	76.50 ± 2.88	8.53 ± 3.00	Acetylcholinesterase inhibitors (36/40)
	40 AD ^2^ (CG ^10^)					16/40	76.33 ± 3.83	8.12 ± 2.79	Acetylcholinesterase inhibitors (38/40)
Martínez-Morenoetal. (2016) [42]	60 AD ^2^	Probable AD ^2^ (mild stage)	NINCDS-ADRDA ^8^	-	Mild stage	25/60	75 ± 6.35 Range = 58–92	Type 1 ^11^ = 37 (62%) Type 2 ^12^ = 13 (21%) Type 3 ^13^ = 7 (12%) Type 4 ^14^ = 1 (2%) Type 5 ^15^ = 0 (0%) Type 6 ^16^ = 2 (3%)	No ChEIs ^17^ = 27 (45%) ChEIs ^17^ = 33 (55%)
Nousia et al. (2018) [38]	25AD ^2^ (TG ^9^)	Mild (early stage) AD ^2^	NINCDS-ADRDA ^8^	1	2.40 (±1.61)	9/25	76.24 (±5.14)	8.08 (±3.01)	-
	25 AD ^2^ (CG ^10^)				3.28 (±2.30)	5/25	76.32 (±5.38)	8.92 (±2.83)	-
Bajpai et al. (2020) [43]	15 AD ^2^	Early AD ^2^	NINCDS ^8^	1	≤8	9/15	60–69: 4/15 (26.7%) 70–79: 7/15 (46.7%) 80–89: 4/15 (26.7%)	0–5: 0/15 (0.0%) 6–9: 1/15 (6.7%) 10–12: 5/15 (33.3%) ≥13: 9/15 (60.0%)	-
Parlak et al. (2023) [39]	16 AD ^2^ (TG ^9^)	6 mild, 6 moderate, 4 severe	DSM-5 ^18^ & NIA-AA ^19^	-	-	7/16	75.00 ± 6.38	3.19 ± 2.90	Acetylcholinesterase inhibitors for at least 3 months
	16 AD ^2^ (CG ^10^)	6 mild, 6 moderate, 4 severe				6/16	74.63 ± 6.60	3.19 ± 2.31	

^1^ N = Number; ^2^ AD = Alzheimer’s disease; ^3^ CDR = Clinical Dementia Rating; ^4^ GDS = Geriatric Depression Scale; ^5^ SD = Standard Deviation; ^6^ LT = Lexical Therapy; ^7^ OT = Occupational Therapy; ^8^ NINCDS-ADRDA = National Institute of Neurological and Communicative Disorders and Stroke and the Alzheimer’s Disease and Related Disorders Association; ^9^ TG = training group; ^10^ CG = control group; ^11^ Type 1 = incomplete primary education; ^12^ Type 2 = complete primary education; ^13^ Type 3 = secondary education; ^14^ Type 4 = vocational education and training; ^15^ Type 5 = undergraduate degree; ^16^ Type 6 = university degree (bachelor’s degree); ^17^ ChEIs = cholinesterase inhibitors; ^18^ DSM-5 = Diagnostic and Statistical Manual for Mental Disorders, fifth edition; ^19^ NIA-AA = National Institute on Aging and Alzheimer’s Association.

**Table 3 healthcare-12-00741-t003:** Summary of the procedural characteristics and quality of life findings of the retrieved studies.

Study	Other Cognitive Domains	Language Domains	Duration of Sessions	Quality of Life
Ousset et al. (2002) [36]	-	Lexical Therapy (naming sessions)	5 months—45 min/session (i) 8 sessions (one session per week) (ii) 2 weeks off (iii) 8 sessions (one session per week)	-
Noonan et al. (2012) [40]	-	Name relearning	10 sessions (participants were seen twice a week over a period of 5 weeks), each lasting between 40 and 60 min	-
Spironelli et al. (2013) [41]	Spatial and temporal orientation, attention, memory, logic reasoning, praxis and arithmetic skill	Language	2 h/day and 4 days/week for 5 weeks + daily living activities (answering a phone call and remembering the message, or reading the newspaper and commenting the news of the day)	The experience of working together encouraged the sense of responsibility of patients with higher cognitive functioning for supporting those with more severe deficits when all participants carried out different everyday activities and tasks
Cavallo et al. (2016) [37]	Memory, attention, executive function	Language	Three 30 min sessions per week, for 12 weeks	Τhe comparison of patients’ scores on the HADS ^1^ did not show any statistically significant difference (anxiety: patients’ score = 7.65 ± 2.41, controls’ score = 7.57 ± 1.33; depression: patients’ score = 6.42 ± 2.21, controls’ score = 6.35 ± 2.21)
Martínez-Moreno et al. (2016) [42]	Reality orientation, memory, executive functions, activities of daily living training	Language (1. discussing actual information of interest, 2. tasks involving reading, oral, and written comprehension and writing, 3. communication between participants)	1 year group program (10–12 patients per group) of two to three weekly sessions (mean 115 sessions/year) of cognitive stimulation and occupational therapy of 2 or 3 h each	Functional capacity in the follow-up after the treatment showed that Responders had a better performance of IADL ^2^
Nousia et al. (2018) [38]	Episodic and delayed memory, attention, processing speed, and executive function	Morphology, syntax, semantics, naming, verbal fluency, and word recall	15 weeks 2 days/week 60 min/session +extra cognitive and language tasks for practice at home, in a weekly basis	The training group had verbal positive feedback on daily activities and functional communication
Bajpai et al. (2020) [43]	Memory (picture recognition task) Attention (spot the differences task)	Verbal learning task	3 tasks per day (30–45 min) for 8 weeks	-
Parlak et al. (2023) [39]	Orientation, reminiscence, executive functions, short-term memory, attention and visual spatial functions, communication board	Language (functional expressions, naming and and showing what is said)	1 h each day for 3 days a week (app sections) and 30 min. each day for 2 days (reminiscence section) Total 5 days/week for 7 weeks	Statements from patients’ caregivers regarding better functioning in everyday life

^1^ HADS = Hospital Anxiety and Depression Scale; ^2^ IADL = Instrumental Activities of Daily Living.

**Table 4 healthcare-12-00741-t004:** Detailed results.

Study	Outcome Measures			*p* Value				*p* Value
Ousset et al. (2002) [36]		LT group (mean ± SD)			OT group (mean ± SD)			
	Naming Hits	Pre	Post			Pre	Post	
	Narrative LT items	31.6 ± 3.7	33.6 ± 3.1		Narrative LT items	32.5 ± 2.8	31.4 ± 2.9	
	LT items	30.1 ± 3.2	31.7 ± 4		LT i tems	28.1 ± 4	28.9 ± 3.5	
	External items	22.9 ± 6.1	24.6 ± 8		External items	23.7 ± 3.5	21.9 ± 4.9	
	Naming Errors	Pre	Post			Pre	Post	
	Absence of production	14.7 ± 8.1	13.6 ± 8.3		Absence of production	16.7 ± 7	20.5 ± 7.3	
	Semantic errors	14.6 ± 5.1	9.9 ± 4.8		Semantic errors	11.7 ± 5.1	12.4 ± 4.6	
	Perceptual errors	6 ± 4.3	6.5 ± 7.3		Perceptual errors	6.6 ± 4.1	5.1 ± 3.8	
Noonan et al. (2012) [40]		Week 1 post-therapy			Week 5 post-therapy			
	Picture version of the Pyramids and Palm Trees Test	r = 0.81, *p* = 0.01						
	Boston Naming Test	r = 0.67, *p* = 0.071						
	100-item naming test	r = 0.67, *p* = 0.066			r = 0.68, *p* = 0.062			
	64-item word-picture matching task				r = 0.65, *p* = 0.076			
	Forward digit span	r = 0.69, *p* = 0.057						
	Camden Recognition Memory for Faces Test	r = 0.62, *p* = 0.09						
	Elevator Counting with Distraction				r = –0.7, *p* = 0.077			
Spironelli et al. (2013) [41]		Pre-treatment	Post-treatment					
	MMSE	22.09 ± 0.58	21.73 ± 0.69					
	MODA	76.26 ± 1.85	77.62 ± 1.98					
	ENB-2: M. I. -10 s	2.45	2.82					
	ENB-2: S.R.-I.	4.64	5.09					
	ENB-2: Abs.	3.45	4.18	0.05				
	ENB-2: Flu.	7.64	8.40					
	ENB-2: Over. Figure	17.09	17.36					
	RTs for LF words	921.02 ± 56.94 ms	1062.85 ± 67.34 ms					
	RTs for HF words	845.69 ± 48.54 ms	973.00 ± 59.80 ms					
	RTs	1008.83 ± 73.17 ms	757.88 ± 48.78 ms	<0.1				
Cavallo et al. (2016) [37]		TG Pre (mean ± SD)	TG 6-month follow up (mean ± SD)		CG Pre (mean ± SD)	CG 6-month follow up (mean ± SD)		
	MMSE	22.65 ± 1.74	22.32 ± 0.97		23.05 ± 2.44	22.64 ± 0.96		
	DSF	4.85 ± 1.60	5.95 ± 1.80		5.20 ± 1.85	5.18 ± 1.82		
	DSB	3.20 ± 1.26	5.78 ± 1.44		4.10 ± 0.63	4.02 ± 0.88		
	Two-syllables word test	4.80 ± 1.72	6.14 ± 1.42		6.00 ± 2.15	5.05 ± 2.15		
	RBMT (standardized profile score)	8.60 ± 1.12	8.60 ± 1.12		8.80 ± 1.36	8.80 ± 1.36		
	RBMT (story immediate)	6.72 ± 1.09	8.72 ± 1.24		7.04 ± 1.66	6.00 ± 1.41		
	RBMT (story delayed)	5.35 ± 1.73	6.35 ± 1.73		6.52 ± 1.66	4.52 ± 1.44		
	GNT	21.95 ± 2.57	22.04 ± 2.53		22.15 ± 2.17	22.18 ± 2.27		
	Token test	30.30 ± 2.42	32.30 ± 2.42		30.69 ± 2.10	27.69 ± 2.10		
	VOSP (object decision)	18.20 ± 0.72	18.25 ± 0.93		18.42 ± 0.81	18.45 ± 0.81		
	VOSP (position discrimination)	19.22 ± 0.70	19.15 ± 0.74		19.29 ± 0.72	19.22 ± 0.70		
	VOSP (number location)	8.87 ± 0.69	8.85 ± 0.58		9.00 ± 0.68	9.02 ± 0.62		
	Verbal fluency (letters)	35.88 ± 2.66	36.57 ± 2.46		36.52 ± 2.45	37.35 ± 2.26		
	Verbal fluency (category)	17.10 ± 1.88	16.27 ± 1.71		17.27 ± 1.76	15.95 ± 1.60		
	Hayling test (overall score)	5.82 ± 1.24	5.42 ± 0.98		5.95 ± 1.15	5.37 ± 0.86		
	Brixton test	4.95 ± 0.85	5.95 ± 1.34		5.22 ± 1.32	3.82 ± 1.65		
Martínez-Moreno et al. (2016) [42]		R-Pre (mean ± SD)	NR-Pre (mean ± SD)		R-Post (mean ± SD)	NR-Post (mean ± SD)		
	Person orientation	55.76 ± 15.31	49.04 ± 18.92	0.16	54.55 ± 16.71	46.66 ± 18.18		0.086
	Space orientation	46.52 ± 17.02	41.76 ± 18.16	0.33	58.29 ± 13.99	52.28 ± 19.67		0.18
	Time orientation	32.03 ± 16.25	36 ± 19.04	0.41	40.52 ± 20.26	28.31 ± 18.40		0.018
	DSF	49.87 ± 10.12	43.64 ± 8.7	0.015	47.06 ± 9.38	43.81 ± 8.06		0.17
	DSB	46.97 ± 8.99	40.52 ± 10.99	0.018	46.17 ± 8.89	43.73 ± 9.9		0.34
	List learning	27.65 ± 10.86	24.48 ± 9.9	0.28	29.24 ± 11.38	26.72 ± 11.86		0.45
	Story memory	24.44 ± 10.47	19.04 ± 8.23	0.048	23.16 ± 12.13	19.72 ± 9.37		0.27
	List learning free recall	19.64 ± 13.31	19.08 ± 13.68	0.88	16.32 ± 13.23	17.26 ± 14.33		0.81
	List learning recognition	19.04 ± 9.45	15.88 ± 8.92	0.23	21.16 ± 14.63	19.44 ± 11.93		0.65
	Story free recall	14.96 ± 7.58	16.20 ± 8.82	0.60	17 ± 9.48	16.04 ± 8.89		0.72
	Figure free recall	27.96 ± 13.68	22.20 ± 12.92	0.13	25.76 ± 15.65	22.04 ± 14.88		0.40
	Visuoverbal naming	41.76 ± 16.71	31.81 ± 15.95	0.027	39.48 ± 14.82	31.13 ± 15.07		0.056
	Constructional praxis	47 ± 16.31	40.75 ± 17.45	0.20	47.96 ± 14.64	40.42 ± 17.45		0.11
	Category evocation	37.59 ± 9.99	32.12 ± 9.78	0.1	36.79 ± 10.65	30.81 ± 10.96		0.045
	MMSE	22.84 ± 3.37	22.79 ± 4.4	0.96	25.23 ± 3.22	20 ± 4.38		0.001
	BI	95.97 ± 3.96	93.97 ± 7.24	0.19	92.58 ± 7.29	89.66 ± 9.06		0.17
	IADL	5.13 ± 1.67	4.52 ± 1.92	0.19	4.68 ± 1.49	3.72 ± 1.96		0.038
Nousia et al. (2018) [38]		TG—pre	TG—post	*p* value	CG—pre	CG—post	*p* value	*p* value (TG post- CG post)
	Recall	17.44 ± 3.66	18.16 ± 3.48	0.151	16.60 ± 3.26	16.20 ± 2.45	0.33	0.887
	Delayed memory	0.16 ± 0.37	1.20 ± 1.08	≤0.001	0.40 ± 0.50	0.12 ± 0.33	0.08	≤0.001
	Word recognition	18.08 ± 1.32	18.68 ± 1.28	0.028	18.40 ± 1.25	17.96 ± 1.48	0.20	0.008
	BNT	11.84 ± 1.57	13.40 ± 1.04	≤0.001	11.64 ± 1.32	11.40 ± 1.30	0.22	≤0.001
	SF	22.12 ± 6.05	28.16 ± 6.08	≤0.001	23.36 ± 7.44	22.16 ± 6.31	0.13	≤0.001
	CDT	8.96 ± 2.22	10.28 ± 2.59	0.01	9.72 ± 1.93	9.52 ± 1.36	0.24	≤0.001
	DSF	5.48 ± 0.71	6.60 ± 1.35	≤0.001	5.04 ± 0.93	4.88 ± 1.13	0.35	≤0.001
	DSB	3.68 ± 0.75	4.32 ± 0.75	0.001	3.36 ± 0.81	3.32 ± 0.98	0.80	0.004
	TMT A	177.24 ± 45.88	151.80 ± 39.48	≤0.001	177.56 ± 56.02	210.16 ± 66.58	0.01	≤0.001
	TMT B	300 ± 00.00	290.60 ± 24.67	0.017	297.84 ± 10.80	299.00 ± 5.00	0.32	0.003
		Week 1	Week 8					
Bajpai et al. (2020) [43]	Memory	48.5 ± 22.9 s	60.5 ± 21.8 s					
	Attention	216.6 ± 78.2 s	286.8 ± 87.0 s					
	Language	211.8 ± 68.4 s	270.4 ± 104.9 s					
Parlak et al. (2023) [39]		TG pre (mean ± SD)	TG post (mean ± SD)	*p* value (TG pre-TG post)	CG pre (mean ± SD)	CG post (mean ± SD)	*p* value (GG pre-CG post)	*p* value (TG -CG post)
	MMSE							
	Orientation	5.19 ± 1.97	6.81 ± 2.48	<0.001	4.56 ± 2.15	4.25 ± 2.11	0.096	0.004
	Registration	2.50 ± 0.89	2.81 ± 0.54	0.136	2.38 ± 1.08	2.69 ± 0.60	0.173	0.542
	Attention and Calculation	1.81 ± 1.90	2.50 ± 2.36	0.007	2.75 ± 1.98	2.25 ± 1.94	0.015	0.747
	Recall	0.20 ± 0.41	0.20 ± 0.41	1.000	0.25 ± 0.57	0.13 ± 0.50	0.164	0.654
	Language	5.69 ± 2.024	7.25 ± 1.06	0.001	6.19 ± 1.37	6.00 ± 1.50	0.383	0.011
	Total	15.38 ± 5.80	19.56 ± 5.76	<0.001	16.13 ± 5.65	15.19 ± 5.46	0.055	0.035
	LATA							
	Speech fluency	23.69 ± 5.33	26.06 ± 4.69	<0.001	24.38 ± 7.38	22.81 ± 7.54	0.001	0.154
	Auditory comprehension	43.81 ± 17.10	52.81 ± 14.88	<0.001	44.88 ± 13.58	44.31 ± 11.97	0.771	0.085
	Repetition	13.38 ± 4.74	16.00 ± 4.14	0.001	14.81 ± 4.94	13.56 ± 3.03	0.083	0.067
	Naming	34.44 ± 11.62	37.38 ± 9.59	0.009	33.25 ± 8.50	32.63 ± 7.88	0.574	0.136
	Reading	31.36 ± 13.60	34.73 ± 13.52	0.007	32.91 ± 14.50	30.82 ± 15.05	0.006	0.529
	Grammar	12.25 ± 4.83	15.06 ± 5.06	<0.001	11.56 ± 5.27	10.31 ± 3.96	0.091	0.006
	Word actions	15.50 ± 5.06	17.69 ± 4.42	<0.001	16.00 ± 4.39	15.13 ± 3.96	0.084	0.095
	Writing	22.73 ± 16.52	25.73 ± 15.08	0.081	25.00 ± 14.58	24.73 ± 14.67	0.341	0.876
	Total	198.63 ± 76.11	223.18 ± 70.43	0.001	211.72 ± 68.83	203.54 ± 58.59	0.210	0.485

LT = Lexical Therapy; SD = Standard Deviation; OT = Occupational Therapy; EL = Errorless Learning; EF = Errorful Learning; AD = Alzheimer’s Disease; MMSE = Mini Mental State Examination; MODA = Milan Overall Dementia Assessment; ENB-2 = EsameNeuropsicologico Breve-2; M.I.-10 s = memory with interference-10 s; S.R.-I. = story recall-immediate; Abs. = abstract verbal reasoning; Flu. = phonemic verbal fluency; Over. Figure = overlapping figure test; RTs = response times; LF = low frequency; HF = high frequency; ms = milliseconds; TG = Training Group; CG = Control Group; DSF = digit span forward; DSB = digit span backward; RBMT = Rivermead Behavioral Memory Test; GNT = Graded Naming Test; VOSP = Visual Object and Space Perception battery; R = Responders after treatment; NR = Non Responders after treatment; BI = Barthel Index; IADL = Instrumental Activities of Daily Living; BNT = Boston Naming Test; SF = Semantic Fluency; CDT = Clock-drawing test; TMT A = trail-making test A; TMT B = trail-making test B; s = seconds; LATA = Language Assessment Test for Aphasia.

## Data Availability

Not applicable.

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
