# Peer review of "Language and Communication Interventions in People with Alzheimer’s Disease: A Systematic Review"

_healthcare, 2024, doi:10.3390/healthcare12070741_

Round 1

Reviewer 1 Report

Comments and Suggestions for Authors

This is a nice and relevant systematic review on a really important topic that affects patients, caregivers and health staff all over the world. As you mention, the language domain has been frequently neglected in the overall design of rehabilitation strategies in AD patients. 

I should recommend to address some particular points:

Have you linked your SR protocol to a Cochrane or a Prospero or any other platform with a registered number? Editors and readers will be ready to use AMSTAR or ROBIS to evaluate the SR, and will be glad to find this topic mentioned.

Search strategy, Prisma flow diagram and data extraction are clean and clear. But you could include in your elegibility criteria no language restriction (english, french, spanish and so on).

Some suggestions for discussion: In the introductory text, it would be useful to discuss how to isolate or split "only language domain".  This is not a problem of the SR. It is a problem of global cognition and language-driven mental skills. Language is a complex non homogenous ability. Fluency, naming and comprehension demand different instruments, even in short evaluation batteries. Comprehension and fluency also open the door for memory evaluation. A brief commentary on these problems is advisable. 

Risk of Bias assessment table is robust. Caution calls in discussion section  can be supported in the two domains showing high risk scores (numbers 2 and 3). In Noussia paper, there are differences in GDS between the groups, whereas in Bajpai an interaction effect seems to be present. I suggest to discuss these results to offer the reader a broader spectrum for interpretation of results. 

Outcome assessment are easy to follow, but late-evoked potentials (N400) distract the real outcome! It is not brain plasticity the focus. It is the clinical benefit of intervention.

Finally.  Do interventions produce stable benefits in the follow-up? Probably this information is not available and is not included in the SR protocol, but must be mentioned as a sensible aspect in AD rehabilitation strategies.

Author Response

Thank you for your comments and suggestions. Attached you will find a point-by-point response.

Reviewer 2 Report

Comments and Suggestions for Authors

General Comments:

The manuscript presents a comprehensive systematic review of language and communication interventions in patients with Alzheimer's Disease (AD). The authors have undertaken a critical analysis of both randomized controlled trials and observational studies, offering valuable insights into the effectiveness of these interventions. This review presents comprehensive research, certain areas require more rigorous scrutiny and clarification to enhance the manuscript's contribution to the field.

Major Points:

1-    Could the authors detail the process used to resolve discrepancies between reviewers during study selection and data extraction? This information would add methodological transparency and strengthen the study's validity.

2-    The discussion section, while informative, could be expanded to provide a more nuanced analysis. Can the authors discuss how the findings align with or differ from existing literature, and what these results suggest about future research directions? A deeper exploration of unaddressed aspects or novel insights offered by the study would be beneficial.

3-    The manuscript mentions the positive impact of language rehabilitation but seems to overlook a critical analysis of why exclusive language rehabilitation shows limited evidence. What might be the underlying factors, and how could future research address these gaps?

Minor Points:

1-    The format of Tables 4 and subsequent in pages 9 and 10 requires adjustments for clarity and readability.

2-    The introduction is comprehensive but could be more concise.

3-    Ensuring that all references are current and pertinent to the topic is crucial.

4-    Considering the study's findings, what specific recommendations can be made for healthcare practitioners regarding the integration of language and communication therapies in AD treatment plans?

The manuscript provides valuable insights into language and communication interventions in AD. Addressing the points raised above, particularly in enhancing methodological transparency and deepening the discussion section, will significantly contribute to its impact and relevance in the field.

Comments on the Quality of English Language

The grammar and sentence structure are coherent, facilitating clear communication of ideas. However, attention to detail in certain sections for grammatical consistency and precision in terminology usage is recommended to enhance the overall readability and professional presentation of the paper.

Author Response

(The authors gave the same response as above.)

Reviewer 3 Report

Comments and Suggestions for Authors

Dear Authors,

1. Kindly use the latest Prisma flow charts for your manuscript. e.g., PRISMA 2020 flow diagram for updated systematic reviews which included searches of databases, registers and other sources 

2. Please rearrange the studies included in a series way or in chronological order.

3. This review lacks proper discussion and evidence for readers to use these interventions for AD.

4. How this manuscript help to reform the clinical practice of AD?

5. Kindly follow proper PRISMA reporting methods for this review.

6. Please add these lines to discussion section___Several limitations need to be acknowledged. These limitations stem, mainly, from 282 the heterogeneity among studies. A methodological limitation is the fact that the partic- 283 ipants in all the included surveys, except for one, were patients with mild AD. Future 284 studies should organize the presentation of their findings based on the severity of the 285 disease (i.e., mild, moderate, or severe) to better reflect the clinical importance of poten- 286 tial benefits at each stage of severity. 287 Moreover, significant inconsistencies are apparent regarding the evaluation tools 288 used in the analyzed articles. The language evaluation should be done with batteries 289 which have good psychometric properties and can investigate various aspects of lan- 290 guage and communication difficulties that pwAD frequently have. According to recent 291 data [29], such tools are the Arizona Battery for Cognitive-Communication Disorders 292 (ABCD) [30], the Sydney Language Battery (SydBat) [31], and the Addenbrooke's Cogni- 293 tive Examination (ACE-III) [32]. Nevertheless, this review includes studies that evaluate 294 Healthcare 2024, 12, x FOR PEER REVIEW 13 of 14 language outcomes with other tools which assay various cognitive domains, including 295 language in general.

7. please revise these suggested changes and edit them in a scientific manuscript. 

Comments on the Quality of English Language

Needs appropriate guidance or editing services.

Author Response

(The authors gave the same response as above.)

Round 2

Reviewer 1 Report

Comments and Suggestions for Authors

I really appreciate the fine considerations to my comments in the first review.

New information included and discussed offered to readers highlight the difficulties in appraising a sistematic review in this fascinating topic. 

No doubt the new text is more robust and informative.  Researchers and readers will welcome this document, and for sure, in the letters to editors en enthusiastic debate will ensue.

Congratulations

Author Response

Thank you for your valuable feedback

Reviewer 2 Report

Comments and Suggestions for Authors

The authors have submitted a revised version of the manuscript, in which the suggestions have been incorporated. I believe that the paper now meets the necessary criteria for publication. 

Author Response

Thank you for your valuable feedback

Reviewer 3 Report

Comments and Suggestions for Authors

Dear Authors,

1. kindly change Fig 1 and only include the included studies in the review. 

2. Kindly explain - Discrepancies were resolved by a 92 third author (I.L.). Also include the interrater reliability for the resolution of discrepancies. Please include a percentage match or agreement.

3. Please shorten the conclusion statements.

4. Include the kind of application for the study included in table 1- Parlak et al. (2023) Turkey AD1 16 TG4 16 CG5 Application at home.

Comments on the Quality of English Language

Mild Improvement needed.

Author Response

Thank you for your valuable feedback
